# Economic Assessment and Control Strategy of Combined Heat and Power Employed in Centralized Domestic Hot Water Systems

**Peijun Zheng** **, Peng Liu \* and Yeqi Zhang**

Yangzhong Intelligent Electrical Institute, North China Electric Power University, Yangzhong 212200, China; zhengpeijun@ncepu.edu.cn (P.Z.); yqzhang@ncepu.edu.cn (Y.Z.)
\* Correspondence: liupeng@ncepu.edu.cn; Tel.: +86-0511-8079-1307

**Abstract:** With the increasing application of CHP and an industry transition to distributed energy, it is necessary to make a comprehensive economic analysis and comparison of the entire lifetime of CHP from the net present value (NPV), payback period, and cost-saving ratio (CSR). Five systems, including micro-CHP, gas boiler (GB), air-source heat pump (ASHP), domestic gas-fired heater and domestic electric hot water-heater, are simulated. First, this paper takes annual heat use efficiency (AHUE) into account to compare the economy of each domestic hot water (DHW) system. The results show that a domestic gas-fired heater system is the most economical option in the AHUE of 31.28%. The economic influence of CHP and gas-fired heater under different AHUE are then analyzed. The results show that the DHW system based on CHP is the best when the AHUE is more than 55.35%. Finally, three different operation strategies of CHP are considered in this paper. From the perspective of annual energy cost and payback, the internal combustion-based CHP with thermal energy system (TES) is superior to the other two strategies being studied. Considering the optimal economic benefits, the CSR of the three different operation strategies is 41.3%, 69.69% and 69.77%, respectively.

**Keywords:** economic assessment; optimal control strategy; combine heat and power; domestic hot water systems; annual heat use efficiency

## 1. Introduction

In 2021, China pointed out that it is necessary to make a good commitment to the peak of carbon emission around 2030 and carbon neutrality by 2060. This means that China's energy mix will change in the future, with 68% thermal power generation and 38% of its total carbon emissions. Therefore, finding a way to generate electricity and heat to limit the development of thermal power generation has become the goal of decarburization in China in the future. Owing to the following characteristics of distributed CHP systems, the government attaches great importance to it:

- Micro-CHP systems are typically installed near the end-user (e.g., commercial or residential buildings) that enable the recycling of waste heat [1], which helps to reduce transportation and distribution losses.
- More than 85% of energy efficiency, low carbide and harmful gas emission, and environmental friendliness.
- Peak load shifting for smart grids and gas pipelines with better economic benefits.

In recent years, great efforts have been made by many researchers on the economic and environmental performance of local micro-CHP systems [2,3]. They point out that the actual economic convenience of a micro-CHP plant strongly depends on the electrical and thermal demand of the user, the electricity and fuel tariffs, the existing incentives, and the proper size of the prime mover (PM) and the TES system. However, it is unpopular to use micro-CHP with higher energy efficiency and low emissions in buildings for the following reasons:

- Higher investment costs compared to DHW systems with low-capacity boilers is not conducive to the spread of the technology.
- China's previous energy mix, management system and electricity/gas price made it difficult to popularize it on a large scale.
- Due to the operating environment and load of CHP system in different regions, the reasonable configuration [2] of Micro-CHP system has a limiting factor for its development.

Since micro-CHP systems with a larger amount of total annual cost saving have a shorter payback period of investment, the investment costs cannot completely reflect advantages and disadvantages of each system [3]. Moreover, because the characteristic of residential is relatively decentralized, the feasibility of the micro-CHP system for residential is a subject worth studying. Many scholars have demonstrated the economic and environmental potential of micro-CHP for residential [4,5]. They compared micro-CHP for residential with a large heating grid or other large heating equipment. However, there is no literature that compares the economization of micro-CHP for residential with the most commonly used domestic heating equipment, such as the domestic electric water-heater and the domestic gas-fired water-heater. Whether the micro-CHP system for residential is better than the domestic heating equipment as the heating method for residential in the future is a problem worth studying. Therefore, it is necessary to analyze its cost-effectiveness. Meanwhile, this work entails an energy and economic study of several water heating options for buildings, with the primary aim to assess the effectiveness of a micro-CHP approach to providing domestic hot water. In this paper, annual heat use efficiency (AHUE) is taken as one of the factors to consider the economic evaluation of DHW system, and the influence of different AHUE on the payback period of micro-CHP investment is analyzed. For a DHW system, AHUE indirectly represents the idleness situation of heating units and can more intuitively measure heat demand. It can also be used as a reference for DHW system configuration.

The rest of this paper is organized as follows. In Section 2, the arrangement of five configurations is introduced. In addition, the modeling of each configuration is proposed. Moreover, an economic assessment is introduced to calculate NPV, payback period and CSR. In Section 3, a case study is used to examine the best arrangement of a distributed DHW system. Optimization results will be compared in Section 4. Finally, this paper is concluded in Section 5.

## 2. Literature Review

Previous studies have demonstrated the economic and the environmental potential of micro-CHP for residential. Hosseinian [4] designed a CHP system which relies on specifications of all equipment, maps, dimensions and equipment layout to prove the feasibility of using CHP in the factories of a manufacturing company in the Tehran Special Economic Zone, Tehran. The company reduced the total cost by 39.07% in comparison with the generation of electricity and heat separately. Asaee et al. [5] took into the simulation that micro-CHP-based Canadian Hybrid Residential End-Use Energy and GHG Emissions Model (CHREM) retrofit yields 13% energy savings and the total GHG emissions of the CHS would be reduced by 35%. This study indicates that ICE-based CHP system retrofits may provide an economically feasible opportunity.

The micro-CHP systems have an advantage of total costs. Meanwhile, the micro-CHP system allows for a reduction of the equivalent carbon dioxide emissions and a reduction of the primary energy consumption in comparison to the conventional system. Qiu [6] applied a genetic algorithm based on optimization strategy of high-temperature PEMFC micro-CHP systems to place emphasis on generating power. The thermal efficiency and total system efficiencies are about 50% and 91%, respectively. A micro-CHP system integrated with geothermal-assisted methanol reformed and incorporated a PEMFC stack [7] such that the annual total cost-saving ratio can reach 20.9% considering methanol price, hydrogen price and service life. It shows that the micro-CHP achieves thermodynamic and economic potential, and provides an alternative way for efficiently using clean energy resources. [8,9]

made a technical and economic analysis on the micro-CHP system of PEMFC and solid oxide fuel cell (SOFC) for single-family residence. The results pointed out that the total efficiency of the SOFC system is 6% higher than that of the PEMFC system. However, from an economic point of view, the payback period of the SOFC system investment is longer than that of PEMFC system in economic terms because its investment cost is not competitive. Finally, it is concluded that the KW-level PEMFC micro-CHP system is suitable for residential that consumes more electricity and heat. The internal combustion-based micro-CHP application in residential buildings in Korea has an advantage in terms of energy-saving and CO2 reduction compared to the baseline model with grid power and a gas-fired boiler systems [10]. However, the price difference between electricity and gas is a hurdle to implement for economic feasibility of micro-CHP applied in residential. In China, gas prices have an advantage when electricity prices peak using power generated by micro-CHP systems to replace electricity bought from the grid [4]. Yang et al. [11] provided sensitivity analysis of the internal combustion-based CCHP system with biomass and gas fuel. They show that compared with biomass prices, natural gas prices have a greater impact on the exergy cost per unit product.

The configuration of micro-CHP is an area that needs to be considered in terms of the economy. It is necessary to find the optimal design of power generators and thermal storage to minimize daily management costs and environmental impact. Meybodi and Behnia [12] proposed a thermal economy model for the effect of carbon tax, which is used to select the appropriate ICE for the CHP system. The economy of micro-CHP also needs to focus on the importance of correctly sizing and designing a micro-CHP residential installation including the TES system, engine power, and the demand for DHW accumulation [2]. Moreover, a reasonable solution has been complemented with the economic yield each micro-CHP provides. It avoids energy waste reduction and saves daily energy cost. Its economic value depends on the NPV and payback period [3].

Jialong Wang [13] studied the impact of ICE capacity at different working hours on the payback period of the CCHP system. They concluded that if the engine capacity reaches more than 100 kW, the payback period will be less than 4 years. Mohammad Mahdi Balakheli [14] analyzed different arrangements of ICE-based CHP systems from energy, economic and environmental viewpoints, and evaluated primary energy savings and net present value, which is achieved in the range of 31% and 36%, respectively. The proposed arrangement has a longer backpack period than other arrangements and higher cost of fuel consumption. Arabkoohsar [15] proposed an innovative waste-CHP-ORC plant to consider two alternative working fluids environmentally friendly for increasing energy and exergy efficiencies by up to 20% and 10%, respectively, and decreasing the payback period of the parallelization project by about 10% from 7.4 years to 6.7 years. It can be seen that internal combustion-based micro-CHP application in residential buildings in Korea has an advantage in terms of energy-saving and $CO_2$ reduction compared to the baseline model with grid power and a gas-fired boiler system [10].

The variations in fuel prices and equipment availability hours also affect the economics of the system. The CHP system for hospitals is most economically profitable when the engine operates 8000 h/year. The total annual energy cost has been reduced by 32.4% [16]. The cost–benefit ratio is greater than one, the net present value (NPV) is positive and the internal rate return for the 20-year lifetime of the system is 19%. Additionally, the annual primary energy consumption has been reduced by 28%, and the annual reduction of pollutant emissions has also been significantly reduced. This paper will continue to analyze the influence of the heat use rate (the use hours of equipment) on the system. However, no literature has considered domestic appliance in residential to compare its economics. Therefore, in this paper, five heating systems, including micro-CHP, ASHP, gas boiler, domestic gas-fired water-heater, and domestic electric water-heater, were compared to find the cost-optimal heating system for residential.

## 3. System Structure

Building energy analysis is conducted by comparing traditional DHW system (GB), CHP system, ASHP system, domestic gas-fired heater system and domestic electrical hot water-heater system. The capacity of each system is calculated based on building structures. District DHW system was modeled as a linear programming (LP) problem [17]. In this section, different models of the DHW are presented, and the aim is to examine the potential for using each DHW and provide a sensitivity analysis.

The studied case is in Yangzhong, China (119.80 N, 32.24 E). The selected building is a scientific research office building in Yangzhong, China. It is composed of five floors of about 5500 m$^2$ a total, 58 rooms, and hosts 100–200 people, including the dormitory, kitchen, laundry, and washroom, etc. Each room hosts 1–4 people. The reason this building was selected as the building is not only the different application scenarios such as office, hotel, residential, etc., but also it is a micro-community that can be simulated, which meets all the requirements of the study. Meanwhile, the heat output of the system includes DHW demand and heat losses of pipeline and insulation. Therefore, heat losses need to be considered.

### 3.1. Each System Configuration

Five systems, including domestic electric heater, domestic gas-fired heater, micro-CHP, ASHP and a traditional DHW system (GB), are compared in this paper. As for heat loads, the DHW in each room is supplied by a domestic electric heater or a domestic gas-fired heater, so the heat loss can be neglected.

The insulation is connected to indoor cool and heat pipes to form a closed loop in each room. The heat loss is considered in the DHW system with micro-CHP and ASHP at the same time. The reason for selecting a domestic electric heater or a domestic gas-fired heater is that they are the most general heaters in residential buildings. Next, micro-CHP and ASHP are selected as heating units to replace the traditional DHW system and assess the economics. The electrification of heat can be achieved through traditional direct electrical heating, or via ASHP. The higher efficiency of heat pumps over direct electrical heating and their lower carbon emissions make heat pumps more attractive. Energy hub and microgrids with CHP is the self-reliance, which consists of maximizing the use of locally generated energy and minimizing the import of energy from the external [18]. In terms of each system component, micro-CHP systems is made of five main components: an internal combustion engine, a TES, an EMS and an electrothermal tube, which converts electricity into heat during low electricity price. Power and heat energy are provided by internal combustion engines. TES is used to balance peak heat output and consumption under all strategies. The excess heat energy is stored in TES to meet the load at the lowest cost, and the peak power of the heat energy is transferred to achieve the peak energy dispatch. EMS controls the working state of heating units. The heating units reach the switch-off situation when the water temperature reaches the designated value ($T_{max}$), while it works at the switch-on temperature ($T_{min}$) where the building gains and losses are balanced, and the DHW is turned on. An electrothermal tube is turned on during the period of off-peak electricity prices, converting the electric energy into heat energy and storing it in TES.

#### 3.1.1. Domestic Electric Heater and Domestic Gas-Fired Heater

We assume that the DHW system is equipped with a domestic electric heater or a domestic gas-fired heater. A domestic electric heater is simple in structure, which includes an electrothermal tube as the main heating unit, but it has the high cost of electricity consumption. Meanwhile, a domestic gas-fired heater has higher technical requirements than domestic electric heaters, resulting in higher installation costs, but lower running costs. The main technical specifications of the domestic gas-fired heater [19] are summarized in Table 1. The corresponding power is thus described by Equation (12). The main technical specifications of the domestic gas heater [19] are summarized in Table 1. The corresponding power is thus described by Equation (12).

**Table 1.** The parameters of domestic gas-fired heater.

| Technical | Unit | Value |
|---|---|---|
| Rated power output (minimum-maximum) | kW | 7–8.5 |
| Flow rate | L/min | 2–11 |
| Gas consumption (LHV standard) | MPa | 0.02–0.8 |
| Efficiency at nominal load | % | 90 |
| Indoor sound power level | dB (A) | 69 |
| Dimensions (height × width × depth) | mm | 240 × 82 × 400 |
| Approximate weight | kg | 3.4 |

### 3.1.2. Air-Source Heat Pump Model

ASHPs extract energy from the external air. The air/water system uses a hydronic system to distribute the heat bin wall radiators or underfloor pipes. Air/air pumps distribute the heat energy through the building via ducts. Carroll [20] has summarized the heat pump-related literature. Heat pumps can be used to meet the DHW demand. However, air-to-source heat pumps (ASHPs) are considered for the following reasons. ASHPs can be easily installed in houses, and they can provide DHW simultaneously. They have a small ground area requirement and are practical in densely populated urban areas [21], so it seems to be the preferred heating technology for the future. The differential equation describing the relevant cost in ASHP refer to the previous research contribution [22]. The corresponding power is thus described by Equation (12).

### 3.1.3. Micro-CHP Model

In the arrangement with the CHP units, the internal combustion engines operate on what is the four-stroke cycle. If the waste heat in the hot water tank cannot meet the heat load of the user, the internal combustion engine will provide a heat source to supplement the heat demand of the user. The mode of the CHP system is complicated, and therefore the energy flow diagram is used to a brief description of the whole operation mode of the system. In the process of energy conversion of thermal system, energy and mass are conserved. Figure 1 as the energy flow diagram can directly show the perfection of heat energy flow, transfer and transformation in the whole process, and grasp the characteristics of energy transformation according to energy distribution, to find ways to improve energy use rate. The design performance of the CHP was analyzed by referencing the datasheet of a report published by the manufacturer [23]. The main design parameters are shown in Table 2. For the micro-CHP system, the energy production is primarily planned based on the heat demand as shown in Equation (12), because the lack of heat is not permitted, and surplus heat must usually be disposed of at a cost, even though it can be partly stored in the TES. The method of energy storage is different from electric storage [24,25] The power of internal combustion engine and the heat stored in TES is defined by Equations (12) and (18).

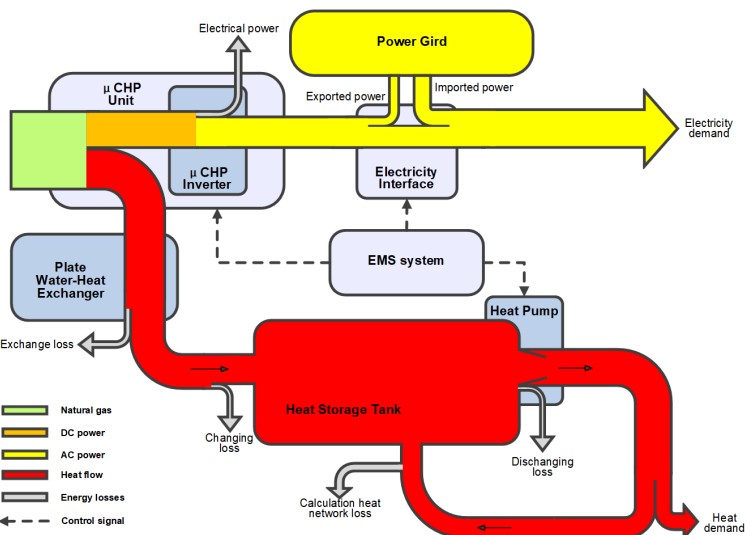

**Figure 1.** The CHP system flow chart.

**Table 2.** The parameters of the internal combustion engine.

| Method | Unit | Value |
|---|---|---|
| Rated power output | kW | 25 |
| Rated power output | kW | 25 |
| Heat power output | kW | 38.5 |
| Gas consumption output | kW | 74.6 |
| Power generation efficiency | % | 33.5 |
| Heat efficiency | % | 51.5 |
| Comprehensive efficiency | % | 85 |
| Frequency | Hz | 50 |
| Voltage | V | AC400 |
| Maximum temperature of water inlet | °C | 85 |
| Water inlet flow | L/min | 110 |
| Supply pressure for fuel gas | kPA | 1.5–3.0 |
| Dimensions (height × width × depth) | mm | 2060 × 2150 × 800 |
| Ambient temperature | °C | −5 °C–40 °C |
| Relative humidity | % | 80 or less |
| Altitude | m | 2000 or less |
| Low heating value (LHV) | MJ/Nm$^3$ | 35.588 |
| Generator Type | | Compact and light weight permanent magnet generator |
| Grid connection method | | High efficiency inverter grid connection |
| Output heat exchanger | | Stainless steel with brazed plate construction |

### 3.2. Calculation Methods

The relevant index about economic saving is one of the most important factors in the investigation of the DHW system. The economic assessment provides information on how the economic resources (investments, fuels, etc.) are used to meet the customer requirements. Compared with traditional energy acquisition methods, micro-CHP system has usually higher initial investment and lower running costs compared with the conventional energy supply system, which serves the electricity load by utility grid and thermal load by gas boiler. Therefore, in economic evaluation, an important index, cost-saving ratio [23], has been employed. The index takes into account more factors. It expresses the profitability of the micro-CHP system and is defined as the ratio of difference between the new system and the conventional system (GB) to the annual energy cost of the conventional system, as shown in (1). Its size indicates the cost-saving capacity of the CHP system.

$$CSR = (\frac{C_{CON} - C_i}{C_{CON}}) \times 100\% \tag{1}$$

$$NPV = -C_i^{initial} + \sum_{j=1}^{n} \frac{C_i^{Save}}{(1+\alpha)^j} \tag{2}$$

$$payback = \frac{C_i^{initial}}{C_i^{Save}} \tag{3}$$

where CSR is cost-saving ratio, $C_{CON}$ and $C_i$ are the annual average investment costs for GB and other systems, $C_i^{initial}$ and $C_i^{Save}$ are initial investment the annual saving cost each system. NPV is net present value. The payback period is the ratio of the initial investment to the amount of cost saved.

$$C_{CON} = C_{CON}^{Inv} + C_{CON}^{Gas} + C_{CON}^{OM} \tag{4}$$

$$C_{CON}^{Inv} = \frac{C_{Boiler}^{initial} \times (1+\alpha)}{n_{CON}} \tag{5}$$

where $C_{CON}^{initial}$, $C_{CON}^{Inv}$, $C_{CON}^{Gas}$ and $C_{CON}^{OM}$ are respectively initial investment, annualized investment cost, annualized gas consumption cost and annualized operation and maintenance cost, for GB. $n_{CON}$ is the lifetime of GB, $\alpha$ is an estimated net residual value rate, j is the number of years.

$$C_i = C_i^{Inv} + C_i^{Gas,or,Ele} + C_i^{OM} - C_i^{in} \tag{6}$$

$$C_i^{Inv} = \frac{C_i^{initial} \times (1+\alpha)}{n_i} \tag{7}$$

$$C_i^{Ele} = \sum_{t=1}^{t=8760} Q_{i,t} \times P_t^{Ele} \tag{8}$$

$$C_i^{Gas} = \frac{\sum_{t=1}^{t=8760} Q_{i,t} \times P_t^{Gas}}{LHV} \tag{9}$$

$$C_i^{in} = \frac{Q_i \times \eta_{i,e-h} \times P_t^{grid}}{3.6} \tag{10}$$

$$C_i^{Save} = C_{CON} - C_i \tag{11}$$

where $C_i^{initial}$, $C_i^{Inv}$, $C_i^{Gas}$, $C_i^{Ele}$, $C_i^{OM}$, $C_i^{Save}$, $C_i^{in}$ and $Q_i$ are respectively initial investment, annualized investment cost, annualized gas consumption cost, annualized electricity consumption cost, annualized operation maintenance cost, annualized saving cost, annualized electricity revenue from grid and actual heat supplied per hour, for one of the CHP, domestic gas-fired heater, domestic electric heater and ASHP. $n_i$ is the lifetime of one of those. $P_t^{Gas}$ and $P_t^{Ele}$ are natural gas price and electricity price at each moment. It is worth noting that i / *in* domestic gas-fired heater, domestic electric heater, micro-CHP and ASHP.

$$Q_i = \frac{Q_{demand}}{\eta_i^{AHUE}} = \frac{E_i}{\eta_i} \tag{12}$$

$$\eta_i^{AHUE} = \frac{Q_i - Q^{loss}}{Q_i} \tag{13}$$

$$\eta_i = \frac{E_{i,h}}{E_{i,g}} \tag{14}$$

$$\eta_{i,e-h} = \frac{E_{i,e}}{E_{i,h}} \tag{15}$$

$$Q^{loss} = \beta \times q \times l \tag{16}$$

$$Q_{TES}^{in} = Q_{i=CHP} \times \eta^{in} \tag{17}$$

$$Q_{TES}^{out} = Q_{TES}^{in} \times \eta^{out} \tag{18}$$

where $Q_{demand}$ $Q_{loss}$, $E_i$, $\eta_i^{AHUE}$, $\eta_i$ and $\eta_{i,e-h}$ are respectively hot water demand, heat losses [20], nominal power, AHUE, heat recovery efficiency and electricity-heat efficiency for each system. $q$ is normative values of specific thermal loss for the analyzed area, $\beta$ is dimensionless coefficient of local thermal losses, considering thermal losses of fittings, mainstays and compensators, $l$ is site length. $Q_{TES}^{in}$ and $Q_{TES}^{out}$ is input heat energy and output heat energy in TES, respectively. $\eta^{in}$ and $\eta^{out}$ is the efficiency of input heat energy and output efficiency.

## 4. Simulation Results

### 4.1. Economic Assessment

The simulation results of five systems with different configurations (including ASHPs, GB, CHP, domestic electric heater, domestic gas heater) are presented and compared. Gas price is USD 0.539/m$^3$. The time-of-use electricity price is implemented in China. Peak electricity prices, flat electricity prices, and off-peak electricity prices used in the baseline scenario are USD 0.167/kWh, USD 0.101/kWh, and USD 0.048/kWh, respectively. To provide a realistic comparison, the heat loss and AHUE of each configuration is considered. Importantly, each proposed DHW is based on comparison with the same hot water demand. Moreover, each DHW system operates at rated power, with the same hot water temperature and no change in heat flow.

Figure 2 shows the high initial investment with CHP. The highest initial investment of CHP remains a large obstacle to applying residential than other configuration of home appliance. Whether it is centralized DHW or distributed DHW, CHP has no advantages on the initial investment. The calculation of investment of different DHW system is from Equation (7).

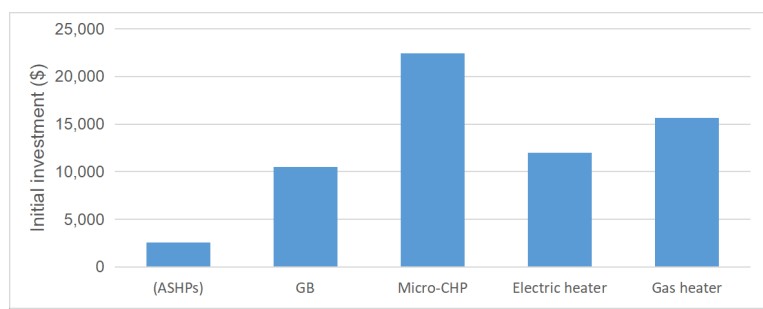

**Figure 2.** The investment of different DHW system

Meanwhile, the DHW system, GB and CHP, refers to the heat transmission and distribution system with more end-users. Use the transmission and distribution pipe network to connect heat users and heat sources to meet local hot water demand. The DHW system, which includes ASHP, domestic gas-fired heater and domestic electric hot water-heater, refers to a heating way by home appliance to meet domestic hot water demand. The DHW system requires longer pipe and consumes more energy than a DHW system in the same demand for hot water. However, the operation costs of a DHW system are low. This paper analyzed the economy of various energy systems. AHUE describes the actually usable and flows into pipeline, taking into account the heat loss of pipeline and gas combustion losses. Through the simulation by the obtained heat data, the AHUE of the CHP system is 31.84% with the heat output efficiency and power output efficiency of 51.5% and 33.5%. The AHUE comparison with different system is shown in Figure 3, which are calculated by Equation (13). The main reason is that the end-users of the selected buildings stay in the building for a short time, the DHW load of the building does not take large portion. In simulation, the heat output is calculated by the same hot water demand and AHUE of each configuration. Figure 4 represented the monthly hot water demand. Energy consumption of hot water in winter season is greater than that in summer season, because

the supply tap water temperature and outdoor air temperature [26] changes according to the seasonal temperature. The lower heat load in February is due to the Chinese New Year, as most people had left the building.

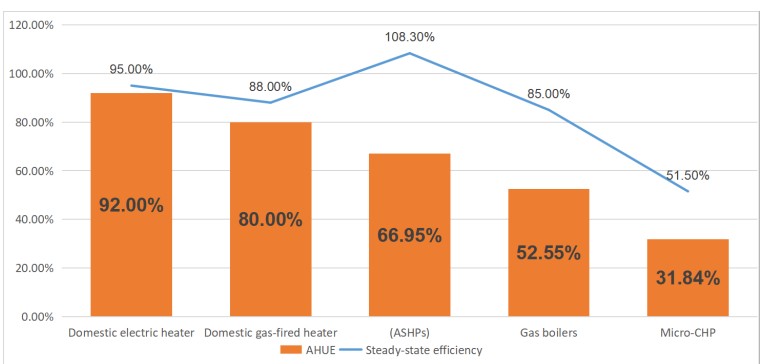

**Figure 3.** AHUE and steady-state efficiency in DHW system of the building.

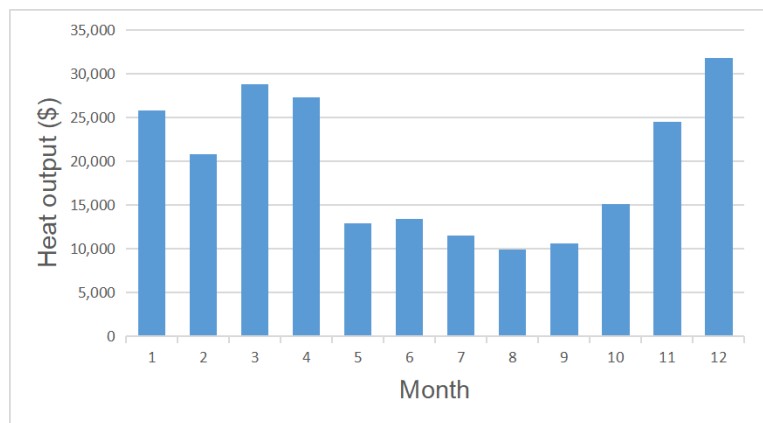

**Figure 4.** Monthly hot water demand with building.

Figure 5 shows the annual energy consumption for each configuration. The annual energy consumption and power saving cost are calculated by Equations (8) and (11), respectively. The micro-CHP, and gas heater is supplied by natural gas, and the rest of the equipment is supplied by electricity. The annual investment cost of different DHW systems is calculated by Equation (7), according to initial investment, interest rate and engine lifetime to eliminate the impact of lifetime. The annual operation of domestic electric heater is too high compared with the traditional system (GB). However, the domestic gas heater on DHW system and CHP on DHW system seems to be an alternative from an economic point of view as well as an engineering one.

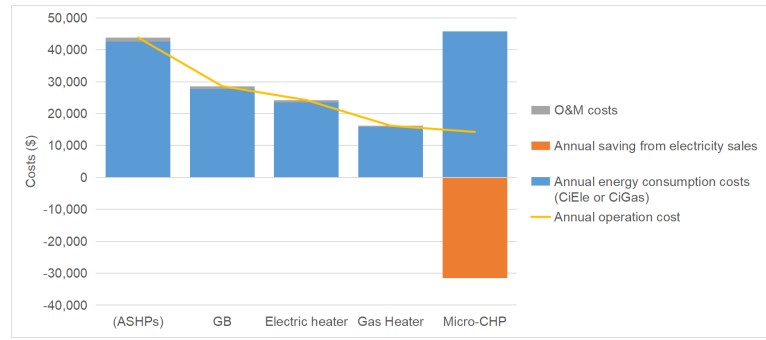

**Figure 5.** Annual energy costs and power saving cost with different DHW system.

Domestic natural gas and CHP have good benefits compared to traditional systems from an annual operating costs point of view, as shown in Figure 5. The annual operating costs of domestic gas heater and CHP consists of O&M costs and total energy costs, which are respectively *USD*2354.43 and *USD*2133.15. Compared with traditional DHW systems, gas-fired heater is saved annually *USD*1201.22 for simulated residential building, and CHP saved annually is *USD*1493.13. However, it is impossible to identify which of the CHP and domestic gas heater is more advantageous without the initial investment. To eliminate the impact of the initial investment of the DHW system for economy, the following further economic analysis between GB and CHP in the traditional DHW system with the instantaneous domestic gas-fired heater with distributed DHW is undertaken.

It is assumed that the hot water and heating temperature range between gas heater and CHP are the same. The payback and CSR are calculated in Table 3, and NPV is shown in Figure 6. The payback of CHP with engine lifetime of 20 is calculated is 9.43 and domestic gas heater with engine lifetime of 8 is 3.55. There is not much benefit of using the CHP from an economic point of view. The main reason is the large heat loss, and the CHP is in standby mode most of the time. AHUE is 31.28%, which means the hot water using time on CHP is too low. Therefore, the economics of micro-CHP are highly dependent on AHUE and are not solely dependent on NPV and payback periods as described in the literature [3]. The heat energy produced in the heating pipes cannot be entirely consumed by the users. When there is no heat demand, some heat energy stored in the pipeline will be released into the air. It causes pipe heat loss in the hot water systems. The influence of the AHUE on CHP payback is discussed below. Figure 7 shows that CHP payback is better than a domestic gas heater when the AHUE reaches 55.35%, which means the average daily operation of CHP units is more than 5.11 h. The payback of is dependent on the sizing of CHP, and the capacity of hot water demand. It is a reference for sizing and designing a CHP residential installation. The ratio will be shorter if the lifetime of each engine is dimensionless.

**Table 3.** CSR and payback for CHP and domestic gas-fired heater.

| Model | CHP | Domestic Gas-Fired Heater |
|---|---|---|
| CSR | 40.11% | 20.34% |
| payback | 9.42 | 3.55 |

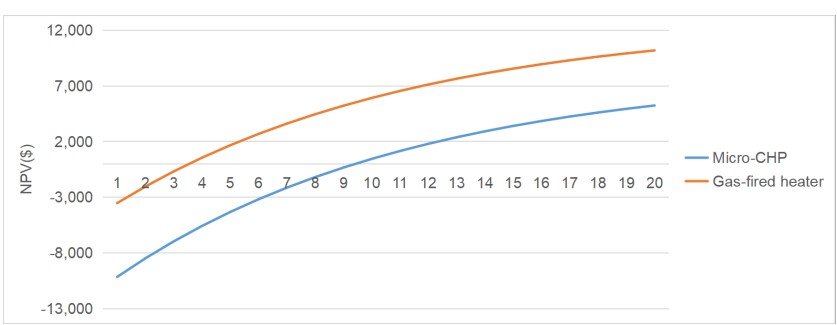

**Figure 6.** Comparison of NPV for CHP and domestic gas-fired heater.

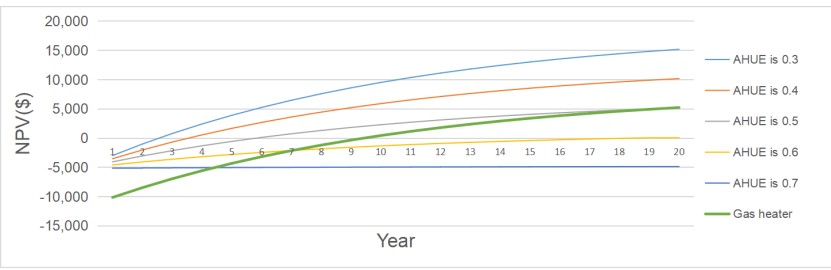

**Figure 7.** Comparison of NPV of CHP for different AHUE.

### 4.2. Control Strategy about Micro-CHP

The previous analysis is based on the same hot water supply temperature between CHP and other configurations. The following studies the optimal economic strategy of micro-CHP, with aim of finding the best threshold by CHP operating at different hot water supply temperatures. When the TES's water temperature is lower than minimum temperature ($T_{min}$), the CHP unit starts to work automatically under the control of the EMS system, and when the water temperature is higher than maximum temperature ($T_{max}$), the CHP unit stops working. It is different from electric-lead strategy [27]. Heat-lead strategy adopts a greedy control strategy in this paper. It means that users can be merely energy consumers. Each user just participates in the use of heat energy, rather than the scheduling of heat energy. Meanwhile, the traditional real-time scheduling control strategy is usually used to optimize the system. The number of feasible decision sequences increases exponentially with the increase of state variables, decision variables, and length of time, which leads to more complex calculation [28]. The advantage of the greedy control strategy is that it is simple to implement and does not require real-time scheduling like electric-lead does. The simulation is still based on the AHUE for 31.28%. The economic efficiency of the three strategies ($S_{chp,1}$, $S_{chp,2}$, and $S_{chp,d}$) is compared with an internal combustion-based CHP system with generating power of 25.3 kW and heating power of 38.5 kW, a TES for heat storage, an EMS system, and plate water-heat exchanger, etc. Considering the different electricity prices at each period, the three strategies correspond with different temperature range and configuration, as shown in Table 4. $S_{chp,d}$ is similar to $S_{chp,2}$, but the difference for both is that during the period of flat electricity prices, the $S_{chp,d}$ system uses electric heating tubes to convert all the electrical energy generated by CHP into heat to reduce the work of CHP. Therefore, the thresholds for starting and stopping the equipment are in different electricity price periods, including off-peak electricity price, flat electricity price, and peak electricity price, which are divided into three intervals. For mode, it is suggested that the start-up temperature should be higher during the high electricity prices. Similarly, it is suggested to reduce the start-up temperature to avoid the operation of CHP units when the electricity prices is low. The above operational logic has been implemented in a home-made numerical code, written in Matlab, which has been realized to establish for each strategy the optimal operation schedule that maximizes the revenues for the CHP concerning the separate generation. Table 5 shows the comparison of annual operating energy consumption, annual revenue, capital saving rate and payback. The NPV of micro-CHP system shows similar results of literature [13].

**Table 4.** The operation temperature range of three control strategies each period.

| Period | $S_{chp,1}$ | $S_{chp,2}$ | $S_{chp,d}$ |
|---|---|---|---|
| 08:00–12:00 and 17:00–21:00 | 39–45 °C | 39–50 °C | 39–50 °C |
| 12:00–17:00 and 21:00–24:00 | 39–45 °C | 39–45 °C | 39–45 °C |
| 00:00–08:00 | 39–45 °C | 35–39 °C | 35–39 °C |

**Table 5.** The economic assessment index for three strategies and traditional system.

| Model | GB | $S_{chp,1}$ | $S_{chp,2}$ | $S_{chp,d}$ |
|---|---|---|---|---|
| Total initial investment | 10,467.29 | 22,429.91 | 22,429.91 | 23,551.4 |
| Annual gas consumption costs | 3403.49 | 6851.42 | 6875.56 | 6670.36 |
| Annual operating profit from electricity sales | – | 4657.87 | 6404.63 | 6466.74 |
| Annual investment costs | 1373.83 | 1177.57 | 1177.57 | 1236.45 |
| Total annual energy costs | 3403.49 | 2193.55 | 470.93 | 203.62 |
| CSR% | – | 40.11 | 70.16 | 73.94 |
| payback | – | 9.42 | 4.13 | 4.23 |

As shown in Figure 8, $S_chp_2$ and $S_chp_d$ have similar NPV. Over time, $S_chp_d$ has a greater NPV than $S_chp_2$. Moreover, Table 5 shows the payback of micro-CHP is approximately 4 years, which is similar results of literature [13]. Compared with the traditional system, the savings of the three strategies increased by $USD4657.87$, $USD6404.63$, and $USD6466.74$ respectively, which means that the annual energy operating cost is reduced by $USD2050.64$, $USD3622.29$, and $USD3686.02$ respectively. The DHW system with electrothermal tube ($S_chp_d$) both power and natural gas while other systems only consume natural gas. Moreover, the CSR of $S_chp_d$ is also higher than other systems. Detailed energy cost and saving of CHP were calculated according to the local electricity and gas price. Gas consumed almost the same cost between $S_{chp,1}$ and $S_{chp,2}$, as shown in Figure 9. The total annual expense of $S_chp_d$ is lowest, since this strategy costs lower and has similar savings in Z0. Meanwhile, there are also noticeable differences between $S_{chp,1}$ and $S_{chp,2}$. Compared with $S_{chp,1}$, the annual total cost of $S_{chp,2}$ has been reduced by nearly $USD1722.64$. The payback of $S_{chp,1}$, $S_{chp,2}$, $S_{chp,d}$ are 9.42, 4.13, and 4.23 respectively, as shown in Table 5. The $S_{chp,d}$ presents less annual gas consumption by electrothermal tube. The total cost of $S_{chp,d}$ is lowest, as shown in Figure 9, but it has longer payback than $S_{chp,2}$ due to the influence of the initial investment.

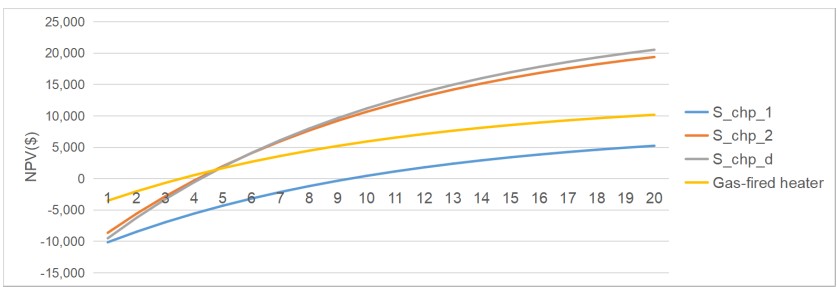

**Figure 8.** Comparison of NPV for CHP in different control strategies and domestic gas-fired heater.

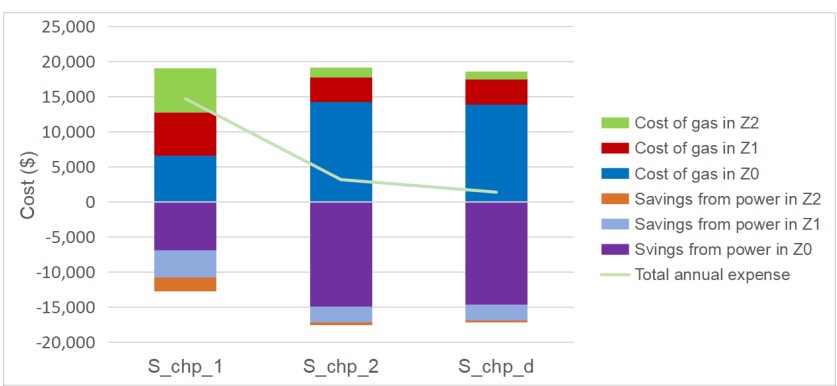

**Figure 9.** Annual energy cost and saving for CHP with different strategies.

Compared with $S_{chp,2}$, the annual savings of $S_{chp,d}$ is almost equal to the annual investment, but it consumes less natural gas, so it has an excellent environmental protection performance. In terms of the savings, the system with electrothermal tubes is higher than other systems. Since CHP with electrothermal tube could further decrease the cost of grid import, the power provided a promising cost-saving potential that may enable a net profit (negative value in Figure 9) for the prosumer when connected with power grid. To sum up, under the condition of meeting the same heat load, the economic benefits of $S_chp_d$ are the strongest.

## 5. Conclusions

As an illustrative example, a typical residential building located in Yangzhong, China, has been examined as a case study. In this paper, the economic performances of the five systems are compared from NPV, payback, and CSR by formulating the evaluation

model for different DHW systems' configuration, including GB, ASHP, domestic gas-fired heater, domestic electric heater, and CHP systems. First, the economic performance of five configurations is compared from the aspects of initial investment, efficiency and annual energy costs. Then, the optimal two configurations of DHW system and DHW system are selected respectively, and the economic performance is further compared with that of traditional DHW system, including payback period, net present value (NPV) and cost-saving ratio (CSR). The results show that the DHW system based on domestic gas-fired heater is the most economical option in the AHUE is 31.28%. The main reason for this type of system is that once the hot water leaves the water storage tank and flows through the system piping, the water temperature drops due to the traveling distance and ambient air temperature, which leads to large amounts of energy being wasted and increases the energy cost. To eliminate the impact of heat use rate on economy, the investment payback period of CHP and domestic gas-fired heaters under different AHUE is analyzed. The results show that the DHW system based on CHP is the best when the ratio is more than 55.35%. Finally, three different operation modes of micro-CHP are considered, including minimum cost operation and minimum energy consumption. The analysis results show that the $S_{chp,2}$ consists of an internal combustion engine, and thermal energy system (TES) is recognized to be a better choice for the examined residential building under investigation from the perspective of economic operation and payback.

The micro-CHP system produces excess electricity that can be sold, the electricity price has a significant impact on the marketization of the micro-CHP system based on internal combustion engines. Meanwhile, specific subsidies for energy savings are adopted by governments. The use of internal combustion engines increases the profitability of CHP when subsidy is available. As a limitation, with the increasing penetration of renewable energy, the price of electricity may further decrease in the future power grid, which weakens the advantages of micro-CHP system, because its economic advantages are realized by saving electricity costs.

Future work will establish a complete evaluation model on AHUE, which can forecast the heat demand of the DHW system at the beginning of system construction. Moreover, it will combine game theory to conduct real-time economic analysis of micro-CHP units to optimize local operating strategies and scheduling strategies.

**Author Contributions:** Conceptualization, P.Z. and P.L.; methodology, P.Z. and P.L.; software, P.Z.; validation, Y.Z. and P.L.; formal analysis, P.Z.; investigation, P.Z.; resources, P.L.; data curation, P.L.; writing—original draft preparation, P.Z.; writing—review and editing, P.Z.; visualization, P.Z.; supervision, P.L. and Y.Z.; project administration, P.L.; funding acquisition, P.L. All authors have read and agreed to the published version of the manuscript.

**Funding:** This work was supported by the National Key R&D Program of China [grant numbers: No.2017YFC0804800]; the Key R&D plan of Jiangsu Province [grant numbers: BE2018111] and the Special Funds for Fundamental Scientific Research Expenses of Central Universities [grant numbers: 2020MS057].

**Institutional Review Board Statement:** Not applicable.

**Informed Consent Statement:** Not applicable.

**Data Availability Statement:** Data is contained within this article.

**Conflicts of Interest:** The authors declare no conflict of interest.

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
