# Peer review of "Economic Assessment and Control Strategy of Combined Heat and Power Employed in Centralized Domestic Hot Water Systems"

_applsci, doi:10.3390/app11104326_

Round 1
Reviewer 1 Report
In my opinion, the paper is in general interesting and nice to read. The manuscript deserves to be published only once the authors fix the following issues.
Literature review
- The main contributions of the paper are clearly described. Nevertheless, from the current manuscript it is not grasp understanding the novelty of the work. The authors should better highlight the innovative aspects of their work in the manuscript.
System design
- The description of the proposed methodology could be deeply improved. First, it could be better to insert at the beginning of the second section (Section 2) an outline about the system scheme/architecture (how many components, the aim of each, etc.); here, a high-level diagram/scheme could also help reader following the whole description. Maybe the placement of Fig. 2 may be anticipated.
- The model and dynamics of CHP and TES are not defined. The author should detail and comment the choice of their model, compared with the other schemes defined in the related literature (e.g., https://doi.org/10.1109/CASE48305.2020.9216875, https://doi.org/10.3390/en12071231 documents that could be cited in the text). The Authors should comment this point.
Problem formulation
- The control strategy used in the energy management of the involved systems strongly affects the results of the analysis. Since the authors select a greedy control strategy, the authors could compare the results of their analysis with the scenario where an advanced and better-performing control strategy (such as model predictive control: e.g., https://doi.org/10.1109/TCST.2021.3056751, https://doi.org/ 10.17775/CSEEJPES.2020.02160, documents that could be cited in the text) is employed. The Authors should comment this point.
- The authors should clarify how they handle the uncertainty of parameters.
Case study
- One of the dimensions that could be taken into account by energy hub and microgrids with CHP is the self-reliance, which consists in maximizing the use of locally generated energy and minimizing the import of energy from the external (e.g., https://doi.org/10.1109/ISGT-Asia.2017.8378441, document that could be cited in the text). The Authors should comment this point.
- There is no sensitivity analysis in the paper. Is it reasonable?
Conclusions
- Conclusions needs to be extended to present further implications for future research and many managerial insights based on the results of the study, as well as limitations.
Minor
- The authors should check that all the used acronyms are explained.
- Mainly the English is good and there are only a few typos. However the paper should be carefully rechecked.
Author Response
I'm glad you commented on my paper. Your questions helped me a lot. Now, I have revised the paper and answered to all your questions. If you have any further questions, please feel free to ask me.

Reviewer 2 Report
Thank you for the opportunity to review this manuscript. The topic addressed in the article is valid. Despite this, the manuscript has several shortcomings outlined in the comments below.
Comment 1: The abbreviation CHP should be given first in the full name and only then should authors use the abbreviation. Likewise, as a keyword I suggest the authors use "Combined Heat and Power" instead of CHP
Comment 2: "Introduction" section, lines 30-31 authors wrote "In recent years, great efforts have been studied by many researchers on the performance of the economic and environmental in the local micro-CHP systems". However, no references were provided. If you state that someone has researched this topic, you should specify who and in what works. References are missing.
Comment 3: Section 1 proposes to split into 2 sections: "Introduction" and "Literature review". The "introduction" section should present the research topic, the current state of research on the topic, the purpose of the research, and the research gap. Whereas in the next section literature review. As it stands, everything is mixed. And what's worse, the research gap is missing. This makes it unclear why this topic is worth researching.
Comment 4: Please explain in more detail, justify why CSR (cost saving ratio) index was used to evaluate project efficiency and is better than other indicators to evaluate project efficiency. Basically, the authors stated that according to [20] it is important in economic evaluation. Such justification is weak.
Comment 5: In the methodological part (section 2) and results (section 3) some elements should be explained more, because the given information is a little bit chaotic. E.g. Fig. 3 (The meteorological data of a typical year in Yangzhong) is presented, but nowhere is it explained why it is presented and the information from this table is not used anywhere (not seen in the body of the article). Section 3 presents results that Section 2 does not mention how they will be calculated. The point is that the reader reading the results does not fully understand why certain results are presented.
Comment 6: The article lacks discussion. True, the authors called section 3 "Simulation result and disscussion", but there is no discussion there. I did not find even one reference of the simulation results to references, results presented in the literature.
Comment 7: Section 4, "Conclusion," did not indicate what the added value of this article was. The purpose of the study stated in Section 4 differs from the paper objective stated in Section 1 (Lines 108-110). Ultimately, it is unclear whether the purpose of the paper was to compare the economic performance of the five systems or to analyze the influence of the heat utilization rate (the utilization hours of equipment) on the system. The two objectives are not the same. In fact, the paper does not clearly state what the aim of the paper is.
Comment 8: Authors wrote in section 4 (Lines 340-341) that "due to the incentive policies issued by the government and gas companies, it can be found that the introduction of CHP system results in large economic benefits". I do not know on what basis the authors make such a conclusion, since in the paper they did not do research on the impact of state policy, nor was there a literature review on this issue.
Author Response
I'm glad you commented on my paper. Your questions helped me a lot. Now, I have revised the paper and answered to all your questions. If you have any further questions, please feel free to ask me. I will revised it carefully.

Round 2
Reviewer 1 Report
Comments to authors
In the revised paper some improvements have been added.
Previous comments and concerns have been sufficiently addressed, except the following one (corresponding to points 3. and 4. of the previous review report).
Specifically, the authors should corroborate the choices in dynamics of storage and loads justifying the made assumptions commenting existing studies:
- The model and dynamics of CHP and TES are not defined. The author should detail and comment the choice of their model, compared with the other schemes defined in the related literature. The Authors should comment for this point for instance:
- Carli et al. "A Robust MPC Energy Scheduling Strategy for Multi-Carrier Microgrids," 2020 IEEE 16th International Conference on Automation Science and Engineering (CASE), Hong Kong, China, 2020, pp. 152-158 (https://ieeexplore.ieee.org/document/9216875).
- Sperstad et al. “Energy Storage Scheduling in Distribution Systems Considering Wind and Photovoltaic Generation Uncertainties”. Energies 2019, 12, 1231 (https://www.mdpi.com/1996-1073/12/7/1231).
- The control strategy used in the energy management of the involved systems strongly affects the results of the analysis. Since the authors select a greedy control strategy, the authors could compare the results of their analysis with the scenario where an advanced and better-performing control strategy (such as model predictive control: e.g., https://doi.org/10.1109/TCST.2021.3056751, https://doi.org/10.17775/CSEEJPES.2020.02160, documents that could be cited in the text) is employed. The Authors should comment for this point for instance:
- Scarabaggio et al. "Distributed Demand Side Management With Stochastic Wind Power Forecasting," in IEEE Transactions on Control Systems Technology (https://ieeexplore.ieee.org/document/9354436).
- Li et al. "A real-time energy management method for electric-hydrogen hybrid energy storage microgrid based on DP-MPC," in CSEE Journal of Power and Energy Systems (https://ieeexplore.ieee.org/document/9265468).
Author Response
I have modified it according to your opinion, please review it
Reviewer 2 Report
Thank you for the opportunity to review this paper again. The work is better written. However, I still have some minor and major comments (because some important comments were almost omitted by the authors).
Comment 1: Abbreviations are still used in the keywords (CHP).
Comment 2: The authors made minimal effort to improve these sections (Introduction and Literature review) by splitting section 1 from the original version into two sections, without making substantive changes. Section 1 'Introduction' still has no research gap or research objective. And the sentence "In order to make end users realise the economic advantages of micro-CHP digitally, it is necessary to analyse its economy" (Lines 47-48) is neither a research objective nor a research gap. The research gap presented by the authors in the Cover letter should also be in the text of the manuscript. Besides, referring to the explanation in the cover letter, I don't know if the authors know what the research gap is in a scientific article. The gap cannot be due to the government work report. Remember that a scientific article is not commissioned by a government agency and is due to a research gap in the literature. The sentence “This work entails an energy and economic study of several water heating options for a building, with the primary aim to assess the effectiveness of a micro-CHP approach to providing domestic hot water” is a research objective and not a research gap.
The structure of the work (Lines 118-122) should be presented in Section 1 and not Section 2.
Comment 3: The explanation of the use of the CSR (cost saving ratio) index to evaluate project efficiency is sufficient and understandable for me. However, I do not understand why the justification (“The economic assessment provides information on how the economic resources (investments, fuels, etc.) are used to meet the customer requirements. As is known to all, micro-CHP system has usually higher initial investment and lower running cost compared with the conventional energy supply system, which serves the electricity load by utility grid and thermal load by gas boiler. Therefore, in economic evaluation, an important index, cost saving ratio, has been employed.”) is not indicated in the revised version of the article. Admittedly, there is a short justification in the text (around Table 2), but I think that including a broader version indicated in the Cover letter would be advisable (because I understand the purpose of the indicator, but the reader reading the article should also be aware of it).
Comment 4: Still the discussion is almost ignored. Admittedly, the authors have discussed the results of the study in section 4 (which is an advantage), but they have not carried out a comparison of these results with the literature (i.e. whether the results of this simulation-study match what the literature has proposed so far or not). I did not find in section 4 or 5 a single reference to the results in the literature (i.e. whether the conclusions are in line with the literature or not).
Author Response

(The authors gave the same response as above.)

Round 3
Reviewer 1 Report
All comments and concerns have been sufficiently addressed. The revised paper deserves to be published.
Author Response
Thank you very much for the reviewers' guidance. I have modified the paper and changed Section 4.2.

Reviewer 2 Report
I am very pleased that the authors have revised this manuscript carefully. Finally, the article has a scientific structure, i.e. it has the research gap presented, the aim of the research, the current state of research on the topic, and the authors' own research. There are still a few minor errors.
Comment 1: Errors due to incorrect translation.
Page 2 Line 50: "However, there is no literature that compares the economics of micro-CHP ...". Rather "economic efficiency" or "economisation".
Page 2 Line 54: "In order to make end users realize the economic advantages of micro-CHP digitally, it is necessary to analyze its economy". Rather "economic efficiency" or "cost-effectiveness" or "cost-efficiency" rather than "economy".
Comment 2: I believe that starting the 'Literature review' section in the following form: "The economic and environmental potential of micro-CHP for residential has been demonstrated " is inappropriate. It gives the impression that the authors have demonstrated and analysed something beforehand. But they have not.
Comment 3: The discussion still needs to be improved. Admittedly, the authors have included elements of it in section 4.2, but this is largely a pretend discussion. An example of this is the last sentence on page 11 (Lines 314-315). The presentation of the research results (Table 5) cannot be regarded as a discussion. Admittedly, section 4.2 deals with the control of the strategy, but the research results are there. In addition, just before Table 5 there is Figure 8, which is not referred to at all in the text.
The discussion should be either after the presentation of results or when discussing the results (but not when presenting new data) either as a separate section or attached to section 5.
Author Response
I am very honored to have the reviewers' guidance on my manuscript and I have made minor modification to the manuscript.
